

# On the spectral depolarisation and lidar ratio of mineral dust provided in the AERONET version 3 inversion product

Sung-Kyun Shin[1], Matthias Tesche[1], Kwanchul Kim[2], Maria Kezoudi[1], Boyan Tatarov[1], Detlef Müller[1], and Youngmin Noh[3]

[1]School of Physics, Astronomy and Mathematics, University of Hertfordshire, United Kingdom
[2]School of Environmental Science and Engineering, Gwangju Institute of Science & Technology (GIST), Republic of Korea
[3]Department of Environmental Engineering, Pukyong National University, Busan, Republic of Korea

**Correspondence:** Youngmin Noh (nym1120@gmail.com)

**Abstract.** Knowledge of the particle lidar ratio ($S_\lambda$) and the particle linear depolarisation ratio ($\delta_\lambda$) for different aerosol types allows for aerosol typing and aerosol-type separation in lidar measurements. Reference values generally originate from dedicated lidar observations but might also be obtained from the inversion of AERONET sun/sky radiometer measurements. This study investigates the consistency of spectral $S_\lambda$ and $\delta_\lambda$ provided in the recently released AERONET version 3 inversion

product for observations of undiluted mineral dust in the vicinity of major deserts: Gobi, Sahara, Arabian, Great Basin and Great Victoria deserts. Pure dust conditions are identified by an Ångstöm exponent $< 0.4$ and a fine-mode fraction $< 0.1$.

The values of spectral $S_\lambda$ are found to vary for the different source regions but generally show an increase with decreasing wavelength. The feature correlates to AERONET retrieving an increase in the imaginary part of the refractive index with decreasing wavelength. The smallest values of $S_\lambda = 35 - 45\,\mathrm{sr}$ are found for mineral dust from the Great Basin desert while

the highest values of 50-70 sr have been inferred from AERONET observations of Saharan dust. Values of $S_\lambda$ at 675, 870, and 1020 nm seem to be in reasonable agreement with available lidar observations while those at 440 nm are up to 10 sr higher than the lidar reference. The spectrum of $\delta_\lambda$ shows a maximum of 0.26-0.31 at 1020 nm and decreasing values as wavelength decreases. AERONET-derived $\delta_\lambda$ at 870 and 1020 nm are close to the lidar reference while values of 0.19-0.24 at 440 nm are smaller than the independent lidar observations. This general behaviour is consistent with earlier studies based on AERONET

version 2 products.

## 1   Introduction

Atmospheric particles interact with the climate system directly by scattering and absorbing radiation and indirectly by affecting cloud formation and evolution (*IPCC*, 2013). In order to quantify the radiative forcing of aerosols on regional and global climate, it is essential to properly estimate the effect of the different aerosol types (*IPCC*, 2013). Mineral dust is one of the most

important aerosol types and is estimated to account for about one third of the global aerosol loading and aerosol optical depth (AOD) (*Kinne et al.*, 2006). Mineral dust affects atmospheric dynamics and chemistry, and exacerbates air quality, visibility, and public health over a broad range of temporal and spatial scales (*Jickells et al.*, 2005; *Morman and Plumlee*, 2013). These effects of mineral dust strongly depend on its size, as well as optical, microphysical and chemical properties. These properties





change when dust is mixed with other aerosol types (*Ginoux et al.*, 2012) whose contributions to a dusty mixture needs to be quantified in order to assess their individual effects.

The extinction-to-backscatter (lidar) ratio ($S_\lambda^p$) and the particle linear depolarisation ratio ($\delta_\lambda^p$) as measured by polarisation-sensitive Raman or high spectral resolution lidar can be used for aerosol classification and to identify the presence of mineral

dust particles in the atmosphere (*Müller et al.*, 2007; *Burton et al.*, 2012, 2013; *Groß et al.*, 2013). The lidar ratio provides insight into the size and absorption of aerosol particles. It is about 20 sr for cloud droplets, increases to values of 40 sr to 60 sr for mineral dust and can be as high as 80 sr for continental pollution (*Müller et al.*, 2007). The particle linear depolarisation ratio is an indicator of particle shape and measurements of this parameter for undiluted dust plumes show values of $\delta_{532}^p = 0.30 - 0.35$. Smaller values indicate a mixture of non-spherical dust with spherical particles such as biomass-burning smoke or anthro-

pogenic pollution (*Freudenthaler et al.*, 2009; *Tesche et al.*, 2009b; *Burton et al.*, 2014, 2015; *Shin et al.*, 2015). Knowledge of $\delta_\lambda^p$ therefore allows for retrieving the contribution of mineral dust to a mixed dust plume by assuming that spherical and non-spherical particles are externally mixed (*Shimizu et al.*, 2004; *Tesche et al.*, 2009b). However, this approach relies on knowing $\delta_\lambda^p$ for different aerosol types in their pure (i.e. undiluted) form. High-quality lidar measurements of undiluted aerosol plumes are the best source for reliable reference values. While such measurements have been conducted for Saharan and Asian dust

(though not for extended periods of time that allow for statistical insight), they are scarce or non-existent for other source regions.

Radiometer measurements within the AErosol RObotic NETwork (AERONET, *Holben et al.* 1998) can provide an alternative for obtaining insight into $S_\lambda^p$ and $\delta_\lambda^p$ for different source regions. AERONET runs automated instruments for direct sun and sky radiation measurements at hundreds of sites. Its unified retrieval provides spectral aerosol optical properties and employs

a spheroid light scattering model (*Dubovik et al.*, 2006) to obtain aerosol microphysical properties such as the particle size distribution and the complex refractive index. Values of $\delta_\lambda^p$ for Saharan and Asian dust as calculated from AERONET data have been presented by *Müller et al.* (2010, 2012) and *Noh et al.* (2017), respectively. These studies found that AERONET-derived dust depolarisation ratios ($\delta_\lambda^p$) at shorter wavelengths are generally smaller than the ones measured with depolarisation lidar while those at 1020 nm resemble lidar-derived values, and thus, could be used to estimate the contribution of mineral

dust to mixed aerosol plumes. AERONET-derived lidar ratios for Saharan dust as presented by *Müller et al.* (2010, 2012) were found to be consistently larger than direct measurements with lidar. *Müller et al.* (2010, 2012) also present a comparison of AERONET-derived $S_\lambda^p$ for Saharan dust to independent lidar observations and show that the values obtained from using the AERONET version 2 inversion are generally larger than the reference values—with $S_{440}^p$ reaching unrealistically high values of more than 80 sr. *Schuster et al.* (2012) combined AERONET measurements with CALIPSO observations to map $S_{532}$ for

dust from sources in northern Africa and on the Arabian peninsula. They found that $S$ decreases from 55 sr at the Atlantic coast to 40 sr in the Middle East – which is in agreement with Raman lidar observations of Saharan (*Tesche et al.*, 2009a, 2011) and Arabian dust (*Mamouri and Ansmann*, 2013) and is the result of a latitudinal changes in the iron content of the dust.

The recently released version 3 of the AERONET retrieval provides $S_\lambda^p$ and $\delta_\lambda^p$ at 440, 675, 870, and 1020 nm as standard inversion products. To our knowledge, this is the first study in which AERONET version 3 lidar and depolarisation ratios

representative for undiluted mineral dust from different deserts are put into the context of available lidar observations. Section 2





introduces the methodology used in this study. In Section 3, we present and discuss our results. A summary of the findings is presented in Section 4.

## 2 Methodology

### 2.1 Theoretical background

Polarisation-sensitive Raman and high spectral resolution lidars are capable of direct measurements of the extinction-to-backscatter (lidar) ratio from the particle backscatter coefficient $\beta_\lambda^p$ and the particle extinction coefficient $\alpha_\lambda^p$ as

$$S_\lambda^p = \frac{\alpha_\lambda^p}{\beta_\lambda^p} \tag{1}$$

as well as of the particle linear depolarisation ratio from measurements of the backscatter coefficient in dedicated depolarisation channels as

$$\delta_\lambda^p = \frac{\beta_\lambda^{p,\perp}}{\beta_\lambda^{p,\parallel}} . \tag{2}$$

The latter parameter requires measuring the return signal in the plane of polarisation perpendicular to that of the emitted polarised laser light and careful calibration of the measurement of the lidar receiver (*Freudenthaler et al.*, 2009; *Freudenthaler*, 2016; *Mattis et al.*, 2009).

AERONET sun/sky radiometers measure direct solar radiation and sky radiation. The measured data are automatically
analysed using the AERONET inversion algorithm (*Dubovik et al.*, 2006). The retrieved aerosol products are available from the AERONET data base (http://aeronet.gsfc.gov/). The recently released version 3 of the AERONET retrieval added spectral particle linear depolarisation ratios and lidar ratios to the list of standard inversion products such as the particle size distribution, the complex refractive index and the single-scattering albedo of the observed particles.

For each observation, the elements $F_{11}(\lambda)$ and $F_{22}(\lambda)$ of the Müller scattering matrix (*Bohren and Huffman*, 1983) are
computed from the particle size distribution and the refractive index that have been inferred from the AERONET inversion product. The element $F_{11}(\lambda)$ is proportional to the flux of scattered light in case of unpolarized incident light while $F_{22}(\lambda)$ strongly depends on the angular and spectral distribution of the radiative intensity (*Bohren and Huffman*, 1983) as measured with AERONET's instruments (*Dubovik et al.*, 2006). From the element $F_{11}(\lambda)$ and the concurrently inferred single-scattering albedo ($\omega$) at the scattering angle of $180°$, the lidar ratio can be computed as

$$S_\lambda^p = \frac{4\pi}{\omega_\lambda F_{11}(\lambda, 180°)} . \tag{3}$$

The calculation of the particle linear depolarisation ratio requires knowledge of the elements $F_{11}(\lambda)$ and $F_{22}(\lambda)$ at the scattering angle of $180°$:

$$\delta^p = \frac{1 - F_{22}(\lambda, 180°)/F_{11}(\lambda, 180°)}{1 + F_{22}(\lambda, 180°)/F_{11}(\lambda, 180°)} . \tag{4}$$



The comparison of $S_\lambda^\mathrm{p}$ and $\delta_\lambda^\mathrm{p}$ as measured by lidar and inferred from AERONET observations therefore allows for an assessment of the AERONET light-scattering model. Such studies have been presented by *Müller et al.* (2010, 2012) for observations of fresh Saharan dust during the Saharan Mineral Dust Experiment (SAMUM-1) in Morocco. All parameters discussed in the following are wavelength dependent and refer to mineral dust particles. To improve readability the corresponding indices will

be omitted from here on.

## 2.2 Data selection

The AERONET sites considered in this study are located near or within the Gobi, Saharan, Arabian, Great Basin, Great Victoria, and Kalahari deserts. Figure 1 shows that a total of 28 AERONET sites have been considered as representative for mineral dust due to their location close to Gobi desert (Dalanzadgad (43°N, 104°E), Dunhuang (40°N, 94°E), and Dunhuang_LZU (40°N,

94°E)); Arabian desert (Hamim (22°N, 54°E), Mezaira (23°N, 53°E), Solar_Village (24°N, 46°E), Bahrain (26°N, 50°E), and Shagaya_Park (29°N, 47°E)); Saharan desert (IER_Cinzana (13°N, 5°W), Capo_Verde (16°N, 22°W), DMN_Maine_Soroa (13°N, 12°E), Banizoumbou (13°N, 2°E), Bord_Badji_Mokhtar (21°N, 0°E), Tamamrasset_INM (22°N, 5°E), Izana (28°N, 16°W), Quarzazate (30°N, 6°W), Oukaimeden (31°N, 7°W), Ras_El_Ain (31°N, 7°W), and Saada (31°N, 8°W)); Kalahari (Gobabeb (23°S, 15°E) and Uprington (28°S, 21°E)); Great Basin (Yuma (32°N, 114°W), Maricopa (33°N, 111°W),

White_Sands (32°N, 106°W), Roger_Dry_Lake (34°N, 117°W), and Goldstone (35°N, 116°W)); and Great Victoria (Tinga_Tingana (28°S, 139°E) and Birdsville (25°S, 139°E)). We consider all level 2.0 observations that were available in February 2018, i.e., time series lasting until end of 2016 to mid of 2017 for most stations.

AERONET inversions are only performed of observations with an AOD larger than 0.4 at 440 nm (*Dubovik et al.*, 2006). The available AERONET level 2 version 3 inversion products for the sites listed above have been filtered to assure that the obtained

values of $S$ and $\delta$ are representative for pure dust conditions (i.e. undiluted dust plumes). Such conditions are reflected by a weak spectral dependence of AOD and a minor contribution of fine-mode particles to the volume size distribution. To account for these features, we only consider observations with a 440/870-nm Ångström exponent ($å_{440/870}$) smaller than 0.4 and a fine-mode fraction ($FMF$) of the volume size distribution of less than 0.10 (*Schuster et al.*, 2006).

The total number of available cases in the AERONET level 2.0 version 3 inversion product used in this study are 147

(Gobi), 6435 (Arabian desert), 12324 (Sahara), 88 (Kalahari), 28 (Great Basin), and 44 (Great Victoria). The requirement for pure-dust conditions for the investigation of AERONET-derived lidar parameters decreased the number of suitable cases to 38 (26%), 3556 (55%), 7228 (59%), 0 (0%), 7 (25%), and 16 (36%), respectively. The ratio of pure-dust cases to the total observations is highest for the Saharan desert. Not a single pure-dust case remained for the Kalahari desert and that region could not be considered for further study. The absolute number of pure-dust cases is also low for the Great Basin and Great

Victory deserts, and thus, findings for these regions need to be taken with care. However, lidar observations for dust from these source regions are also scarce which is why the AERONET-derived values might provide a valuable reference for future studies or the analysis of measurements with elastic backscatter lidars which require an *a-priori* estimate of $S$ for the retrieval of backscatter and extinction coefficient profiles.



## 3 Results and Discussion

Table 1 presents the results of our analysis of AERONET version 3 products with respect to pure mineral dust conditions in the form of mean values and standard deviation of the particle linear depolarisation ratio, the lidar ratio, the complex refractive index and the single-scattering albedo at four wavelengths and for the considered regions. The first lines of Table 1 furthermore

provide the mean values of $AOD$ at 675 nm, $å_{440/870}$ and the coarse-mode effective radius found for the different regions as well as the number of available pure-dust cases and the ratio of pure-dust to total observations.

### 3.1 Particle linear depolarisation ratio

Figure 2 shows the AERONET-derived frequency distributions of $\delta$ at 440, 675, 880, and 1020 nm for the Gobi, Arabian and Saharan deserts. The absolute number of cases for the other regions was found too low to obtain reasonable distributions (see

Table 1). The values of spectral $\delta$ vary between 0.15 and 0.36 with slight variations for the considered regions. The maximum of the $\delta$ distribution decreases as the wavelength decreases and is highest for the Saharan desert. The Gobi and Saharan deserts show similar frequency distributions while that for the Arabian desert is shifted to slightly lower values. In comparison to the AERONET/lidar study of $\delta$ for Saharan dust of *Müller et al.* (2012), the AERONET version 3 distribution shows an overall improvement by moving to larger, more realistic values. Nevertheless, $\delta$ for lower wavelengths is likely still too low as will be

discussed in more detail below.

Figure 3 compares the spectral variation of the mean AERONET-derived $\delta$ to spectral depolarisation lidar measurements reported in the literature. A more detailed overview of published values of $\delta$ is provided in Table 2. Lidar measurements of $\delta$ are generally performed at 355 or 532 nm. So far, triple-wavelengths measurements of $\delta$ have only been presented in the studies of *Freudenthaler et al.* (2009); *Burton et al.* (2015) and *Haarig et al.* (2017). The AERONET-derived values for

different regions in Figure 3 peak between 0.27 and 0.31 at 1020 nm and decrease steadily to 0.19 to 0.24 at 440 nm. Lidar measurements show that the spectral dependence of $\delta$ differs with origin and age of the observed dust plume. *Burton et al.* (2015) found a maximum of $\delta_{1064} = 0.38$ and lower values of 0.37 and 0.24 at 532 and 355 nm, respectively, for local North American dust. In contrast with this, Saharan dust which has aged during transport shows a peak of 0.30 at 532 nm with smaller $\delta$ of 0.27 and 0.25 at 1064 and 355 nm, respectively (*Burton et al.*, 2015). *Haarig et al.* (2017) reported a similar pattern in the

spectral variation of $\delta$ for aged Saharan dust with a clear maximum of 0.28 at 532 nm and lower values of 0.25 at 355 nm and 0.23 at 1064 nm. The values for fresh Saharan dust observed by *Freudenthaler et al.* (2009) are almost identical to those of long-range transported Saharan dust by *Burton et al.* (2015). All four scenarios agree on a $\delta_{355}$ of about 0.25.

Further findings of $\delta$ from the literature are summarised in Table 2. Lidar observations of $\delta$ are most commonly performed at 532 nm though there is an increasing number of observations at 355 nm. The literature gives values at 532 nm in the range from

0.30 to 0.35 for Asian dust (*Sugimoto and Lee*, 2006; *Shin et al.*, 2015; *Hofer et al.*, 2017) and slightly lower values for aged Asian dust after intercontinental transport (*Cottle et al.*, 2013). A much broader range of values from 0.18 to 0.29 has been reported at 355 nm for observations of central Asian dust (*Hofer et al.*, 2017). A $\delta_{532}$ of 0.31±0.02 has been reported for pure and aged Saharan dust by *Freudenthaler et al.* (2009), *Veselovskii et al.* (2016), and *Haarig et al.* (2017), respectively. However, the





range of values of $\delta_{532}$ for transported Saharan dust extends from 0.28 (*Preißler et al.*, 2011) to 0.34 (*Wiegner et al.*, 2011) while observations during intense dust episodes at M'Bour, Senegal, showed values of 0.35±0.05 (*Veselovskii et al.*, 2016). Values of between 0.28 and 0.35 have been found at 532 nm for Arabian dust over Cyprus (*Mamouri and Ansmann*, 2017). Measurements at other regions of the globe are scarce and often restricted to aircraft campaigns. For instance, *Burton et al.*

(2015) provide values of 0.33 to 0.37 at 532 nm for freshly emitted North American dust in the Great Basin region.

While AERONET-derived $\delta$ is only available at 440, 675, 880, and 1020 nm compared to the lidar wavelengths of 355, 532, and 1064 nm, Figure 3 shows that its spectral variation does not resemble the one derived from lidar measurements for pure and aged mineral dust – a feature that has already been identified by *Müller et al.* (2010, 2012). However, the spectral resolution of the AERONET observations – with a blind spot at the important lidar wavelength of 532 nm – does not allow for

a fair comparison to the literature values presented above. The AERONET-derived values at the longer wavelengths of 870 and 1020 nm fall well within the envelope of the lidar observations as already noted by *Noh et al.* (2017). This suggests that $\delta_{870}$ or $\delta_{1020}$ might be used to derive the contribution of mineral dust to mixed dust plumes analogous to *Tesche et al.* (2009b) and *Burton et al.* (2014).

## 3.2   Lidar ratio

The AERONET-derived frequency distributions of $S$ at 440, 675, 880, and 1020 nm for the Gobi, Arabian and Saharan deserts are presented in Figure 4. The distributions for the Gobi and Arabian deserts are rather narrow compared to those for Saharan dust. All three regions show a clear shift to larger values at 440 nm as well as a broadening of the distribution. In contrast to the frequency distributions of $\delta$ presented in Figure 2, the lidar ratio shows a stronger regional variation. The difference in the lidar ratio for Saharan and Arabian dust at 532 nm has previously been obtained from combining AERONET and spaceborne

Cloud-Aerosol Lidar with Orthogonal Polarisation (CALIOP) observations (*Schuster et al.*, 2012) as well as from Raman lidar measurements (*Mamouri and Ansmann*, 2013; *Nisantzi et al.*, 2015). A closer look at the spectral variation of the AERONET-derived $S$ is provided in Figure 5 and Table 1. Two features are instantly apparent: Saharan dust shows the highest values of $S$ at any considered wavelength and the values at 440 nm are highest for any considered region. Furthermore, the lowest values are found for the Great Basin and Great Victoria deserts followed by the Arabian and Gobi deserts. Figure 5 also contains the

AERONET-based values presented by *Müller et al.* (2010), the findings of the modelling study of *Gasteiger et al.* (2011), and lidar observations of fresh and aged Saharan dust at 355 nm and 532 nm (*Tesche et al.*, 2009a, 2011; *Groß et al.*, 2015). Further context to the literature on dust lidar ratios as measured with lidar at 355 nm and 532 nm is provided in Table 3. While the table refers to mean values presented in the respective papers, the range of observed $S$ at 355 nm extends to the high values shown in Table 1 and Figure 5. For instance, *Veselovskii et al.* (2016) report that about 10% of lidar ratios observed at 355 nm show

values between 65 and 75 sr. Note that measurements of the lidar ratio of mineral dust at larger wavelengths are not available to date. The comparison to reference lidar measurements in Figure 5 and Table 3 reveal that the focus of observations has so far been on dust from the Sahara as the world's larges dust source. However, mean values for fresh and aged Saharan dust still vary in a range of as much as 8 sr at both 355 nm and 532 nm. While observations of Arabian and Asian dust are becoming more



common, there is no literature values for the Great Basin and Great Victoria deserts. Consequently, the AERONET-derived lidar ratios presented here might provide a reference for future lidar observations of mineral dust from these regions.

The closure studies for Saharan dust presented in *Müller et al.* (2010, 2012) provide an extensive discussion of the limitations of AERONET-derived lidar ratios – which are related mostly to the challenge of properly inferring the imaginary part of the

complex refractive index. Compared to these studies, however, Figure 5 reveals that the AERONET version 3 lidar ratios for Saharan dust moved much closer to the lidar observations at 532 nm as well as to model simulations that apply particle shapes of greater complexity than AERONET's spheroid model (*Dubovik et al.*, 2006; *Gasteiger et al.*, 2011). It therefore seems that AERONET-derived values at 675, 870, and 1020 nm are rather reliable. Lidar ratios at 440 nm on the other hand are likely to exceed the actual values, though no longer as extremely as shown by *Müller et al.* (2010). Further discussion on the likely

source of this overestimation is provided in the next section.

## 3.3 Regional differences in dust mineralogy

Table 1 also presents the spectral complex refractive index and single-scattering albedo for the considered regions. Real parts of the refractive index were found nearly independent of wavelength and vary between 1.45 for Saharan dust and 1.57 for Great Basin dust. The imaginary parts of desert dust as obtained from AERONET version 3 data in this study range between

0.0010 at 675 to 1020 nm to 0.004 at 440 nm. Only the short wavelength shows increased values of the complex refractive index, leading to the increase in the lidar ratio presented in Figure 5. Particularly high values are found from observations representative for Saharan and Great Victoria dust. The resulting spectral variation of the complex refractive index with a sharp drop from 440 nm to 675 nm seems unrealistic when compared to the in-situ measurements presented in the closure studies of *Müller et al.* (2010, 2012). Observations with in-situ instruments show that the spectral slope of the imaginary part is not

nearly as steep as the spectral slope inferred from the AERONET inversion. A single-scattering albedo of 0.97 or 0.98 is found at all wavelengths apart from 440 nm for which it drops to 0.87 to 0.93 depending on the different desert regions.

The variation of the lidar ratio of mineral dust from the Saharan and Arabian deserts has been shown by *Schuster et al.* (2012) and *Nisantzi et al.* (2015). These differences are the result of regional changes in the mineralogical composition of the dust particles. In general, mineral dust consists to varying degrees of, e.g., kaolinite, illite, smectite, vermiculite, calcite, quartz,

chlorite, goehite, feldspars, and hematite, whose properties regarding light scattering and absorbing differ considerably. For instance, quartz has strong light-absorption bands in the infra-red (IR) atmospheric window while its absorption properties are negligible at ultra-violet (UV) and visible wavelength. Clays such as illite, kaolinite, and montmorillonite absorb light at solar wavelength and feature an imaginary part of the refractive index that increases towards shorter wavelengths. Hematite is also known as a strongly light-absorbing material in the UV and visible wavelength regions. In contrast, calcite and gypsum show

almost no absorption thoughout the UV and visible spectrum, but have strong light absorption in the IR (*Sokolik and Toon*, 1999).

Consequently, the light-absorption properties of mineral dust are determined by the relative abundance of the different mineral type. The proportions of each mineral compound are differed for different source regions. An overview of the regional change in the composition of mineral dust from the source regions considered in this study as available in the literature is




provided in Table 4. Clay minerals (e.g., kaolinite, illite, smectite) are most abundant in dust from the Sahara with contributions of 73% to 81% (*Sokolik and Toon*, 1999; *Jeong and Achterberg*, 2014; *Journet et al.*, 2014). In contrast, smaller clay fractions of 45% to 66% were found for source regions of Asian and Arabian dust (*Sokolik and Toon*, 1999; *Jeong*, 2008; *Jeong and Achterberg*, 2014; *Journet et al.*, 2014). As shown in *Schuster et al.* (2012), the larger values of the lidar ratio we

have obtained for Saharan dust (see Table 1 and Figure 5) are the result of a higher proportion of clay in the mineral composition, and thus, an increase in the imaginary part of the complex refractive index compared to less clay-rich dust. On the other hand, dust from Australian deserts consists mostly of quartz with a fraction of 95% to 99.7% (*Muhs*, 2004; *Qin and Mitchell*, 2009). This provides a reasonable explanation for the lower lidar ratios we have obtained for dust from the Great Victoria desert. Dust from the different source regions considered in this study contains some amount of iron oxide minerals (e.g., goethite and

hematite) which are major light absorbers in the short wavelength region. While the strong increase of the imaginary part of the complex refractive index at 440 nm (and the resulting decrease of the single-scattering albedo and increase of the lidar ratio with respect to the longer wavelengths) might not be realistic with regard to its amount (*Müller et al.*, 2010, 2012), the general feature is caused by the spectral light-absorbing characteristics of the iron-related mineral in the dust.

## 4  Summary and Conclusions

In this study we investigated the spectral particle linear depolarisation ratio $\delta$ and the particle lidar ratio $S$ as provided in the recently released version 3 of the AERONET inversion. To select observations representative for pure mineral dust conditions, only AERONET data with a 440/870-nm Ångström exponent below 0.4 and a fine mode fraction below 0.10 have been selected in this study. AERONET stations considered here were chosen according to their location and are assumed to represent observations of mineral dust from the Gobi, Arabian, Saharan, Great Basin, and Great Victoria deserts. No suitable AERONET

cases could be found for the Kalahari desert.

AERONET version 3 data show a spectral dependence of $\delta$ with a maximum between 0.27 and 0.31 at 1020 nm for the different source regions and lower values for smaller wavelengths. The minimum values range between 0.19 and 0.24 at 440 nm. AERONET-derived $\delta$ are generally within the range of independent lidar observations of mineral dust though provided at wavelengths different from those used by advanced depolarisation lidars. The AERONET findings of spectral $S$ show a much

stronger regional variation than those for $\delta$. This is because the lidar ratio depends on the complex refractive index (which is determined by mineral composition) while the depolarisation ratio is more dependent on particle morphology. The lowest values of $S$ are found for mineral dust from the Great Basin and Great Victoria deserts. Note that these two regions provided the smallest sample sizes of this study, and thus, should be taken with care. Most values of $S$ for Arabian dust and Saharan dust are in agreement with the lidar literature – though findings at 440 nm are likely to be somewhat too large when compared

to lidar measurements at 355 nm and 532 nm. Apart from 440 nm, the spectral variation inferred using AERONET's spheroid model resembles that obtained from more complex particle models.

The particle linear depolarisation ratio and the lidar ratio are intensive aerosol parameters that allow for aerosol classification and aerosol-type separation. However, for the latter methodology to be reliable, reference values of $\delta$ for pure mineral dust and





other undiluted aerosol types need to be known for different source regions. Our analysis of AERONET-derived dust-related $\delta$ and $S$ shows that both parameters depends on the source region – though to different degrees. We conclude that AERONET measurements at longer wavelengths can provide reference values of $\delta$ and $S$ for pure dust conditions in regions where direct measurements with advanced depolarisation Raman or high spectral resolution lidar are either not available at all or not yet

5 sufficient to obtain statistically robust results. In addition, the inferred spectral dependence of $\delta$ suggests a pathway for obtaining the columnar contribution of mineral dust and other aerosol types to mixed dust plumes from AERONET-derived $\delta$ analogous to its application in the analysis of depolarisation lidar observations.

*Competing interests.* The authors declare that no competing interests are present.

*Acknowledgements.* We thank the PIs of the AERONET sites used in this study for maintaining their instruments and providing their data to

10 the community. We also like to thank AERONET for their continuous efforts in providing high-quality measurements and derivative products. All data used in this work can be accessed through the AERONET homepage http://aeronet.gsfc.gov/. Dr Sung-Kyun Shin was supported by the Basic Science Research Program of the National Research Foundation of Korea funded by the Ministry of Education under grant NRF-2017R1A6A3A03005398. This work has also been supported by the Korea Meteorological Administration Research and Development program under Grant KMIPA 2015-6150.





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

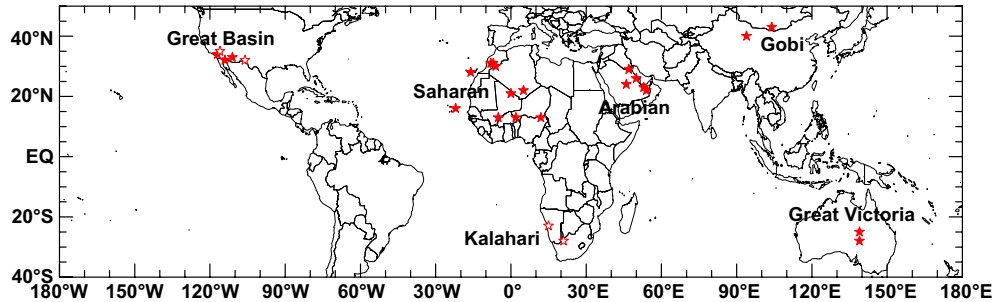

**Figure 1.** Map of the AERONET sites considered in this study. Open symbols mark sites for which the requirement for pure-dust conditions has not been fulfilled.

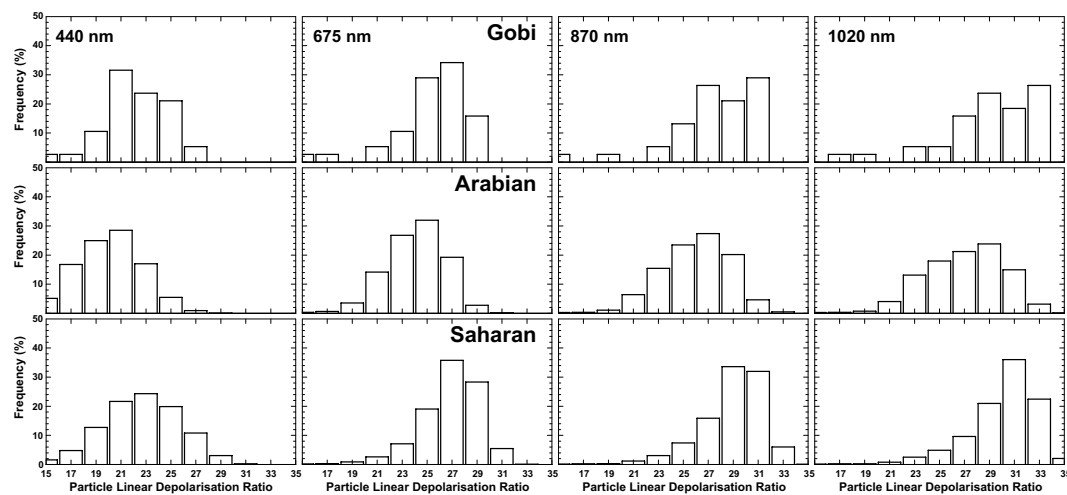

**Figure 2.** Frequency distribution of $\delta$ for pure dust conditions. Rows refer to the Gobi, Arabian, and Saharan deserts while columns refer to the wavelengths of 440, 675, 870, and 1020 nm, respectively. Due to the low number of pure dust cases, Great Basin (N=7) and Great Victoria (N=16) are not shown. Statistics for all regions are provided in Table 1.





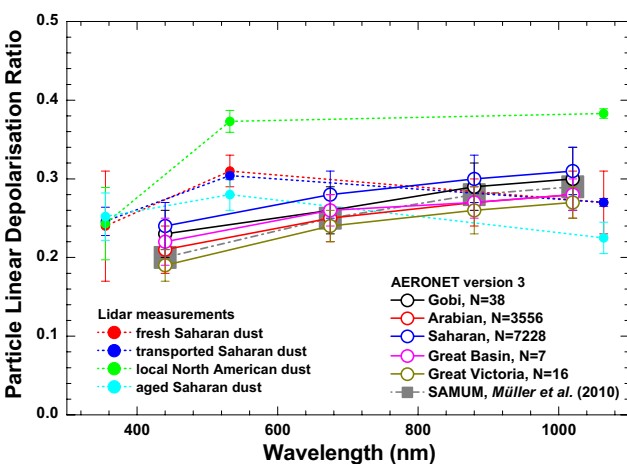

**Figure 3.** Spectral variation of $\delta$ as derived from AERONET observations for pure dust (open symbols) compared to published values from triple-wavelengths lidar observations (solid symbols) of fresh Saharan dust close to the source (*Freudenthaler et al.*, 2009), transported Saharan dust and local North American dust (*Burton et al.*, 2015), and aged Saharan dust after one week of transport (*Haarig et al.*, 2017). The gray line and symbols present the AERONET-derived $\delta$ from the comparison study for Saharan dust of *Müller et al.* (2010).

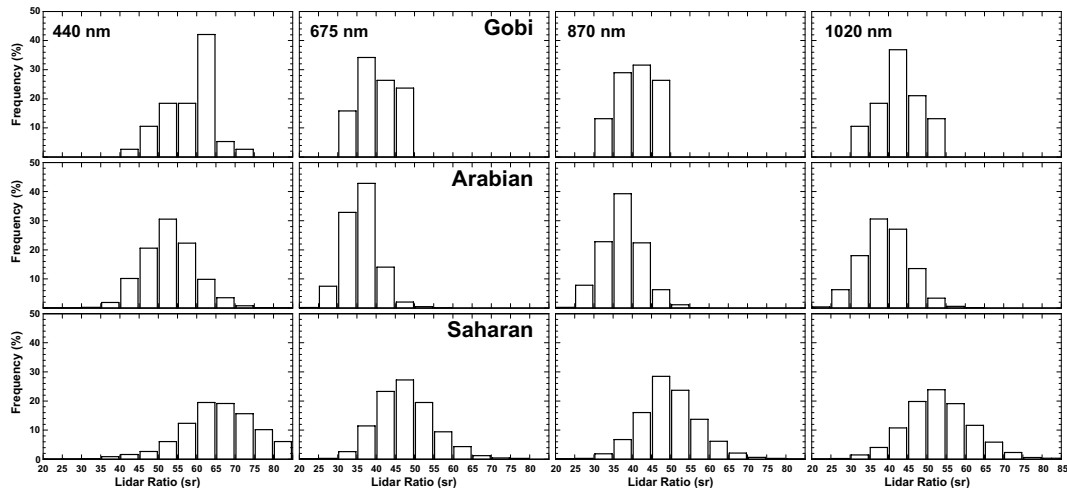

**Figure 4.** Same as Figure 2 but for the dust lidar ratio $S$. Statistics for all regions are provided in Table 1.





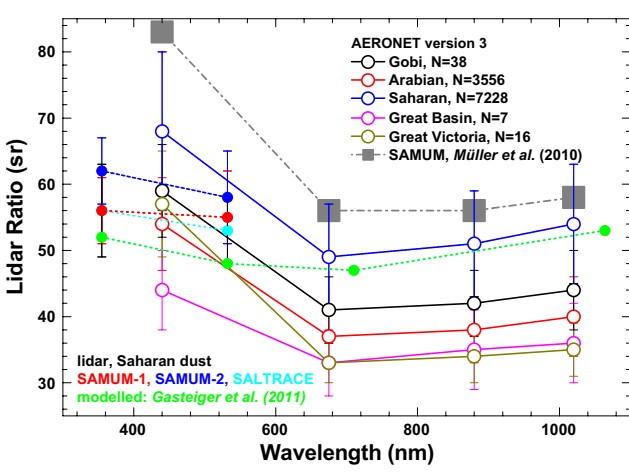

**Figure 5.** Spectral variation of $S$ as derived from AERONET observations for pure dust (open symbols) compared to published values (solid symbols) from two-wavelengths lidar observations of Saharan dust during its transport over the Atlantic (SAMUM-1, *Tesche et al.* 2009a; SAMUM-2, *Tesche et al.* 2011; SALTRACE, *Groß et al.* 2015) and the results of light-scattering modeling (*Gasteiger et al.*, 2011). The gray line and symbols present the AERONET-derived $S$ from the comparison study for Saharan dust of *Müller et al.* (2010).



**Table 1.** Mean values and standard deviation of the AERONET-derived $\delta$, $S$, refractive index ($RI_\mathrm{r}$ and $RI_\mathrm{i}$) and single-scattering albedo ($\omega$) at 440, 675, 870, and 1020 nm together with the number of pure-dust cases (N) and their ratio with respect to all available version 3 level 2 inversion outputs, as well as the average $AOD_{675}$, Ångström exponent ($\mathring{a}_{440/870}$), fine-mode fraction ($FMF$), and coarse-mode effective radius ($r_\mathrm{c}$) for those cases. Note that the requirement for pure dust conditions was not met at the sites representing the Kalahari desert.

| Region | Gobi | Arabian | Saharan | Great Basin | Great Victoria |
|---|---|---|---|---|---|
| N | 38 (26%) | 3556 (55%) | 7228 (59%) | 7 (25%) | 16 (36%) |
| $AOD_{675}$ | 0.85±0.69 | 0.63±0.27 | 0.71±0.34 | 0.44±0.04 | 0.45±0.07 |
| $\mathring{A}_{440/870}$ | 0.16±0.09 | 0.18±0.10 | 0.14±0.07 | 0.08±0.07 | 0.14±0.09 |
| $FMF$ | 0.06±0.02 | 0.07±0.02 | 0.07±0.02 | 0.03±0.01 | 0.05±0.01 |
| $r_\mathrm{c}$ [$\mu$m] | 1.86±0.12 | 1.86±0.17 | 1.78±0.19 | 2.08±0.07 | 2.08±0.23 |
| $\delta_{440}$ | 0.23±0.03 | 0.21±0.03 | 0.24±0.03 | 0.22±0.03 | 0.19±0.02 |
| $\delta_{675}$ | 0.26±0.03 | 0.25±0.03 | 0.28±0.03 | 0.26±0.02 | 0.24±0.02 |
| $\delta_{870}$ | 0.29±0.03 | 0.27±0.03 | 0.30±0.03 | 0.27±0.02 | 0.28±0.03 |
| $\delta_{1020}$ | 0.30±0.04 | 0.28±0.03 | 0.31±0.03 | 0.28±0.02 | 0.27±0.02 |
| $S_{440}$ [sr] | 59±7 | 54±7 | 68±12 | 44±6 | 57±8 |
| $S_{675}$ [sr] | 41±5 | 37±4 | 49±8 | 33±5 | 33±3 |
| $S_{870}$ [sr] | 42±5 | 38±5 | 51±8 | 35±6 | 34±4 |
| $S_{1020}$ [sr] | 44±6 | 40±6 | 54±9 | 36±6 | 35±4 |
| $RI_{\mathrm{r},440}$ | 1.51±0.04 | 1.53±0.04 | 1.47±0.05 | 1.57±0.04 | 1.56±0.04 |
| $RI_{\mathrm{r},675}$ | 1.51±0.04 | 1.54±0.04 | 1.48±0.05 | 1.56±0.04 | 1.56±0.04 |
| $RI_{\mathrm{r},870}$ | 1.49±0.04 | 1.52±0.04 | 1.46±0.04 | 1.54±0.04 | 1.55±0.04 |
| $RI_{\mathrm{r},1020}$ | 1.48±0.05 | 1.50±0.04 | 1.45±0.05 | 1.54±0.04 | 1.53±0.05 |
| $RI_{\mathrm{i},440}$ | 0.0030±0.0011 | 0.0035±0.0012 | 0.0040±0.0023 | 0.0020±0.0006 | 0.0050±0.0020 |
| $RI_{\mathrm{i},675}$ | 0.0009±0.0006 | 0.0010±0.0008 | 0.0012±0.0013 | 0.0007±0.0001 | 0.0011±0.0006 |
| $RI_{\mathrm{i},870}$ | 0.0010±0.0009 | 0.0011±0.0010 | 0.0012±0.0015 | 0.0010±0.0002 | 0.0014±0.0011 |
| $RI_{\mathrm{i},1020}$ | 0.0010±0.0009 | 0.0012±0.0012 | 0.0013±0.0018 | 0.0013±0.0003 | 0.0015±0.0015 |
| $\omega_{440}$ | 0.92±0.02 | 0.91±0.02 | 0.90±0.03 | 0.93±0.02 | 0.87±0.03 |
| $\omega_{675}$ | 0.98±0.01 | 0.98±0.02 | 0.97±0.02 | 0.98±0.01 | 0.97±0.02 |
| $\omega_{870}$ | 0.98±0.02 | 0.98±0.02 | 0.98±0.02 | 0.98±0.01 | 0.97±0.02 |
| $\omega_{1020}$ | 0.98±0.02 | 0.98±0.02 | 0.98±0.02 | 0.97±0.00 | 0.97±0.01 |





**Table 2.** Literature values on lidar observations of the particle linear depolarisation ratio $\delta$ for mineral dust from different source regions. The first column gives the location of the measurements. Note that coincident measurements at three wavelengths are reported only by *Burton et al.* (2015), *Freudenthaler et al.* (2009) and *Haarig et al.* (2017).

| | 355 nm | 532 nm | 1064 nm | Reference |
|---|---|---|---|---|
| **Gobi dust** | | | | |
| Vancouver, Canada | – | 0.27 | – | *Cottle et al.* (2013) |
| Dushanbe, Tajikistan | 0.18-0.29 | 0.31-0.35 | – | *Hofer et al.* (2017) |
| Tsukuba, Japan | – | 0.35 | 0.35 | *Sugimoto and Lee* (2006) |
| Gwangju, South Korea | – | 0.30-0.33 | – | *Shin et al.* (2015) |
| **Arabian dust** | | | | |
| Limassol, Cyprus | – | 0.28-0.35 | – | *Mamouri and Ansmann* (2013) |
| **Saharan dust** | | | | |
| Ouarzazate, Morocco | 0.240±0.007 | 0.310±0.020 | 0.270±0.040 | *Freudenthaler et al.* (2009) |
| M'Bour, Senegal | — | 0.300±0.045 | – | *Veselovskii et al.* (2016) |
| Evora, Portugal | — | 0.280±0.040 | – | *Preißler et al.* (2011) |
| Munich, Germany | 0.290±0.070 | 0.340±0.020 | – | *Wiegner et al.* (2011) |
| Barbados | 0.252±0.030 | 0.280±0.020 | 0.225±0.020 | *Haarig et al.* (2017) |
| Caribbean | – | 0.327±0.018 | 0.278±0.001 | *Burton et al.* (2015) |
| Midwest US | 0.246±0.018 | 0.304±0.005 | 0.270±0.005 | *Burton et al.* (2015) |
| **Great Basin dust** | | | | |
| Pico de Orizaba | – | 0.334±0.018 | 0.400±0.009 | *Burton et al.* (2015) |
| Chihuanhuan Desert | 0.243±0.046 | 0.373±0.014 | 0.383±0.006 | *Burton et al.* (2015) |



**Table 3.** Literature values on lidar observations of lidar ratio $S$ [sr] for mineral dust from different source regions.

|  | 355 nm | 532 nm | 1064 nm | Reference |
|---|---|---|---|---|
| **Gobi dust** | | | | |
| Dushanbe, Tajikistan | 40-47 | 36-43 | – | *Hofer et al.* (2017) |
| Tsukuba, Japan | – | 42-55 | – | *Liu et al.* (2002) |
| **Arabian dust** | | | | |
| Limassol, Cyprus | – | 34-39 | – | *Mamouri and Ansmann* (2013) |
| **Saharan dust** | | | | |
| Ouarzazate, Morocco | 55±5 | 56±5 | 59±7 | *Tesche et al.* (2009a) |
| M'Bour, Senegal | 54±8 | 53±8 | – | *Veselovskii et al.* (2016) |
| Evora, Portugal | 45±11 | 53±7 | – | *Preißler et al.* (2011) |
| Munich, Germany | 58±8 | 61±6 | – | *Wiegner et al.* (2011) |
| Barbados | 53±5 | 56±7 | – | *Groß et al.* (2015) |

**Table 4.** Literature values on the contribution (in %) of clay, quartz and iron in mineral dust from different source regions. The table refers to Kaolinite (Kao), Illite (Ill), Smectite (Sme), Chlorite (Chl), Vermiculite (Ver), Calcite (Cal), Quartz (Qua), Goetheite (Goe), Hematite (Hem), and Feldspars (Fel).

| Desert region | Clays | | | | | Quartz/calcite | | Iron related | | | Reference |
|---|---|---|---|---|---|---|---|---|---|---|---|
|  | Kao | Ill | Sme | Chl | Ver | Cal | Qua | Goe | Hem | Fel |  |
| Asia | 25 | 18 | 15 | 3 | 4 | 6 | 4 | 3 | 1 | 1 | *Journet et al.* (2014) |
| Arabia | 24 | 19 | 17 | 5 | 1 | 9 | 4 | 2 | 1 | 1 | *Journet et al.* (2014) |
| Sahara | 30 | 24 | 20 | 4 | 6 | 9 | 5 | 2 | 1 | 2 | *Journet et al.* (2014) |
| Australia | 29 | 17 | 16 | 3 | 2 | 4 | 3 | 2 | 1 | 0 | *Journet et al.* (2014) |
| Saudi Arabia | 55 | 5 | – | – | – | – | – | – | – | – | *Sokolik and Toon* (1999) |
| Saharan dust | 32 | 41 | – | – | – | – | – | – | – | – | *Sokolik and Toon* (1999) |
| Asian dust | 1 | 19 | 23 | 2 | – | 8 | 28 | – | – | 8 | *Jeong* (2008) |
| Asian dust | 3 | 49 | – | 6 | – | 5 | 16 | – | – | 2 | *Jeong and Achterberg* (2014) |
| Saharan dust | 6 | 72 | – | 3 | – | 2 | 8 | – | 1 | – | *Jeong and Achterberg* (2014) |
| Australia | – | – | – | – | – | – | 99.7 | – | 0.33 | – | *Qin and Mitchell* (2009) |
| Australia | – | – | – | – | – | – | 95–98 | – | – | – | *Muhs* (2004) |
| North America | – | – | – | – | – | – | 83–90 | – | – | – | *Muhs* (2004) |