# Peer review of "On the spectral depolarisation and lidar ratio of mineral dust provided in the AERONET version 3 inversion product"

_Atmospheric Chemistry and Physics, 2018_

## Referee Comment (RC1) · Anonymous Referee #3 · 12 Aug 2018

This article carefully compared the two spectral characteristics of lidar ratio($S_L$) and linear depolarization ratio(delta_L) for the coarse size distribution (AE < 04, and fine mode fraction < 0.1).

As well known for other scientist, these two values depend on the particle morphology(depolarization) and imaginary-refractive index(lidar ratio) of particle. But as author have described in this article at page 2 the line 5-6. Lidar ratio also depend on the particle size. So equation (2) and (3) should be changed from $F_{11}$ ($\lambda$,n=n_r+jn_i) & $F_{22}$ ($\lambda$,n) to $F_{11}$ (x=$2\pi$r/$\lambda$,n) & $F_{22}$ (x,n)(Borhen and Huffman, 1983). So, the author should consider aerosol size distribution for the all kinds of their discussion. When

this size distribution does not changes S_L and delta_L so much, the author should describe this results quatitatively also.

Linear polarization, may be, depend on the size distribution also(because scattering F_11 & F_22 depend on the wavelength and size), for this purpose they must consider this effects and include some results or referenes.

Figure 3, and 5 gives clear spectral changes of lidar ratio and linear depolarization. The author explain these results by using aerosol refractive index( Table 1). I think their explanation is correct. But they did not say anything about the spectral shape of linear depolarization. They must discuss more carefully about this spectral changes. For example, when we consider approximately, that wavelength is longer than aerosol size(x≫1) the morphological shape cannot influence scattering. So, when wavelength increase linear depolarization should decrease. But their results (Figure 3) show opposite picture.

If aerosol linear depolarization ration depend only on the aging period and transportation distance. Please remove line 12-14( "The spectrum of (delta_lambda)_ shows a maximum of 0.26-0.31 at 1020 nm and decreasing values as wavelength decreases. AERONET-derived (delta_lambda)_ _ at 870 and 1020 nm are close to the lidar reference while values of 0.19-0.24 at 440 nm are smaller than the independent lidar observations " at the abstract. So, I think this article can be published in this article when they consider aerosol size distribution in comparing S_L and delta_L

---

## Referee Comment (RC2) · Anonymous Referee #1 · 13 Aug 2018

Authors provide useful comparison of lidar and depolarization ratios of dust provided by AERONET (version 3.0) with corresponding lidar derived values. AERONET results are given at four wavelengths for different source regions. The manuscript is clearly and well written. I wouldn't call the presented results very novel, but it is good overview of lidar and AERONET dust observations, which will be useful for researches working in the field of desert dust study.

I don't have many suggestions for revision, just technical corrections:

Ln 9. "a mixture of non-spherical dust with spherical particles such as biomass-burning smoke" Actually smoke particles are not spherical and in some cases can provide

significant depolarization. So I would use "low-polarizing particles".

Ln 24. "while those at 1020 nm resemble lidar-derived values" At what laser wavelength?

---

## Author Comment (AC1) · 24 Aug 2018

Author replies to the Referee comments on ACP-2018-401: On the spectral depolarisation and lidar ratio of mineral dust provided in the AERONET version 3 inversion product by Shin et al.

**Anonymous Referee #1**

Authors provide useful comparison of lidar and depolarization ratios of dust provided by AERONET (version 3.0) with corresponding lidar derived values. AERONET results are given at four wavelengths for different source regions. The manuscript is clearly and well written. I wouldn't call the presented results very novel, but it is good overview of lidar and AERONET dust observations, which will be useful for researches working in the field of desert dust study.

We thank the Referee for the positive feedback.

I don't have many suggestions for revision, just technical corrections:

Ln 9. "a mixture of non-spherical dust with spherical particles such as biomass-burning smoke" Actually smoke particles are not spherical and in some cases can provide significant depolarization. So I would use "low-polarizing particles".

Thank you for this comment. We have replaced the word "spherical" by "weakly depolarising".

Ln 24. "while those at 1020 nm resemble lidar-derived values" At what laser wavelength?

To clarify our statement, we have revised the sentence to: "…while those at 1020 nm resemble lidar-derived values at 1064 nm (though few observations are currently available at this wavelength), and thus, could be used to estimate the contribution of mineral dust to mixed aerosol plumes."

**Anonymous Referee #3**

This article carefully compared the two spectral characteristics of lidar ratio ($S_\lambda$) and linear depolarization ratio ($\delta_\lambda$) for the coarse size distribution (AE < 0.4, and fine mode fraction < 0.1).

As well known for other scientist, these two values depend on the particle morphology (depolarization) and imaginary-refractive index (lidar ratio) of particle. But as author have described in this article at page 2 the line 5-6. Lidar ratio also depends on the particle size. So equation (2) and (3) should be changed from $F_{11}(n=n\_r+jn\_i)$ & $F_{22}(n)$ to $F_{11}(x=2r/n)$ & $F_{22}(x,n)$ (Bohren and Huffman, 1983).

We thank the reviewer for spotting this oversight on our side. We have replaced all $F_{11}(\lambda)$ and $F_{22}(\lambda)$ through $F_{11,\lambda}(r,n)$ and $F_{22,\lambda}(r,n)$, respectively, in the text and in Eqs. (3) and (4).

So, the author should consider aerosol size distribution for the all kinds of their discussion. When this size distribution does not changes $S_\lambda$ and $\delta_\lambda$ so much, the author should describe this results quantitatively also.

We thank the reviewer for this comment. To investigate the possible effect of changes in the size distributions on our findings, we have looked at the average AERONET-derived size distributions for the intervals of particle linear depolarisation ratio and lidar ratio and used in the histograms in Figures 2 and 4 of the manuscript, respectively. Figures 1 and 2 below show that the average size distributions for intervals of $\delta$ and $S$ that contribute more than 5% to the total data show virtually no

difference for the shape of the particle size distribution for different wavelengths, source regions or intervals of $\delta$ and $S$. This absence of major changes in the shape of the size distribution is likely the result of our initial constraints (AE < 0.4, and FMF < 0.1) for selecting the data suitable for representing pure-dust conditions.

[Figure]

**Figure 1:** Mean AERONET-derived size distributions averaged according to the intervals of $\delta$ used in the histogram in Figure 2 of the original manuscript for intervals that contribute to at least 5% of the data.

[Figure]

**Figure 2:** Mean AERONET-derived size distributions averaged according to the 5-sr intervals of $S$ used in the histogram in Figure 4 of the original manuscript for intervals that contribute to at least 5% of the data.

The following text has been added to the Summary to account for this in the manuscript:
*"To investigate the potential effects of changes in particle size on δ and S, we have obtained the mean size distributions for those intervals in the histograms of Figures 2 and 4, respectively, that contribute more than 5% to the total data (not shown).We found virtually no difference in the shape of the mean size distributions at different wavelengths, source regions, or intervals of δ and S, which is likely the result of our initial constraints for selecting the data suitable for representing pure-dust conditions."*

Linear polarisation, may be, depend on the size distribution also (because scattering $F_{11}$ & $F_{22}$ depend on the wavelength and size), for this purpose they must consider this effects and include some results or references.

We thank the reviewer for this comment. While the reviewer is correct that $F_{11}$ & $F_{22}$ depend on the size parameter (x = 2 \pi r /\lambda), work on modelling intensive lidar parameters of mineral dust by *Wiegner et al.* (2009) and *Gasteiger et al.* (2011) has shown that the complex refractive index and the choice of particle shape (i.e. spheres, spheroids or irregularly shaped particles) has the largest effect on the simulation of *S* and *δ*. However, *Wiegner et al.* (2009) also state that for both quantities, the changes due to size or shape are comparable and significant. The AERONET inversion uses a fixed aspect ratio distribution of spheroids. It is therefore indeed necessary to consider the effect of changes in the particle size distribution. As described in the answer to the previous comment, we have investigated this effect by looking at the mean size distributions for the intervals of *δ* and *S* used in the histograms in Figures 2 and 4 of our manuscript and found no changes in the shape of the size distribution for different wavelengths, source regions or intervals of *δ* and *S*. We therefore conclude that regional changes in the complex refractive index have the largest effect on the obtained values and spectra of *S*.

Figure 3 and 5 gives clear spectral changes of lidar ratio and linear depolarization. The authors explain these results by using aerosol refractive index (Table 1). I think their explanation is correct. But they did not say anything about the spectral shape of linear depolarization. They must discuss more carefully about this spectral changes. For example, when we consider approximately, that wavelength is longer than aerosol size (x»1) the morphological shape cannot influence scattering. So, when wavelength increases linear depolarization should decrease. But their results (Figure 3) show opposite picture.

We had already added the findings of *Müller et al.* (2010) for AERONET observations of Saharan dust during SAMUM as grey squares and lines to Figures 3 and 5 to show that the spectral shape we found for *δ* and *S* for dust from different source regions generally agrees with previous studies. It has been shown by *Wiegner et al.* (2009) that the use of spheroids particles is not ideal for light-scattering simulations of lidar parameters (i.e. to obtain scattering information at 180° backscatter direction) as their solution space only overlaps with that of actual atmospheric measurements if low values of the real part of the refractive index are assumed. *Gasteiger et al.* (2011) showed that lidar measurements of *S* and *δ* for pure mineral dust are best reproduced if the number of particles with aspect ratios below 1.4 is kept small and if irregular particle shapes are used rather than spheroids. Because of these intrinsic limitations of the spheroid model used in the AERONET inversion, we have already written in the conclusions of the original manuscript:
*"We conclude that AERONET measurements at longer wavelengths can provide reference values of δ and S for pure dust conditions in regions where direct measurements with advanced depolarisation Raman or high spectral resolution lidar are either not available at all or the number of observations is not yet large enough to obtain statistically robust results."*
Nevertheless, we have added the following text to the discussion of Figure 3 to better account for the findings of the modelling studies of *Wiegner at al.* (2009) and *Gasteiger et al.* (2011):

*"It has to be noted that light-scattering simulations of lidar parameters using spheroids has its limitations as their solution space only overlaps with that of actual atmospheric measurements if low values of the real part of the refractive index are assumed (Wiegner et al., 2009). In addition, Gasteiger et al. (2011) have shown that lidar measurements of δ for pure mineral dust are best reproduced if the number of particles with aspect ratios below 1.4 is kept small and if irregular particle shapes are used rather than spheroids."*

If aerosol linear depolarization ration depend only on the aging period and transportation distance. Please remove line 12-14 ("The spectrum of $\delta_\lambda$ shows a maximum of 0.26-0.31 at 1020 nm and decreasing values as wavelength decreases. AERONET-derived $\delta_\lambda$ at 870 and 1020 nm are close to the lidar reference while values of 0.19-0.24 at 440 nm are smaller than the independent lidar observations " at the abstract.

We might have misunderstood the Referee's comment but we don't see any reason to remove this statement. It describes exactly what we have found, i.e. the spectral change of $\delta$ presented in Figure 3, and what has been seen in the studies of *Wiegner et al.* (2009) and *Müller et al.* (2010). Nevertheless, we have revised the statement to:
*"The spectrum of $\delta_\lambda$ shows a maximum of 0.26-0.31 at 1020 nm and decreasing values as wavelength decreases. AERONET-derived $\delta_\lambda$ at 870 and 1020 nm are in line with the lidar reference while values of 0.19-0.24 at 440 nm are smaller than the independent lidar observations by a difference of 0.03 to 0.08."*

So, I think this article can be published in this article when they consider aerosol size distribution in comparing $S_\lambda$ and $\delta_\lambda$.

Thank you very much for the positive review and the valuable comments on considering particle size. We hope our answers are to your satisfaction.

**References**

[revised manuscript text omitted]